# Gastric Cancer and Viruses: A Fine Line between Friend or Foe

**DOI:** 10.3390/vaccines10040600

**Published:** 2022-04-13

**Authors:** Ahmad Firoz, Hani Mohammed Ali, Suriya Rehman, Irfan A. Rather

**Affiliations:** 1Department of Biological Sciences, Faculty of Science, King Abdulaziz University, Jeddah 21589, Saudi Arabia; firoz2006@gmail.com (A.F.); haniolfat@hotmail.com (H.M.A.); 2Princess Dr Najla Bint Saud Al-Saud Center for Excellence Research in Biotechnology, King Abdulaziz University, Jeddah 21589, Saudi Arabia; 3Department of Epidemic Disease Research, Institute for Research and Medical Consultation (IRMC), Imam Abdulrahman Bin Faisal University (IAU), Dammam 31441, Saudi Arabia

**Keywords:** Epstein–Barr virus, herpes simplex virus, onco-virus, gastric cancer

## Abstract

Gastric cancer (GC) is a significant health concern worldwide, with a GLOBOCAN estimate of 1.08 million novel cases in 2020. It is the leading cause of disability-adjusted life years lost to cancer, with the fourth most common cancer in males and the fifth most common cancer in females. Strategies are pursued across the globe to prevent gastric cancer progression as a significant fraction of gastric cancers have been linked to various pathogenic (bacterial and viral) infections. Early diagnosis (in Asian countries), and non-invasive and surgical treatments have helped manage this disease with 5-year survival for stage IA and IB tumors ranging between 60% and 80%. However, the most prevalent aggressive stage III gastric tumors undergoing surgery have a lower 5-year survival rate between 18% and 50%. These figures point to a need for more efficient diagnostic and treatment strategies, for which the oncolytic viruses (OVs) appear to have some promise. OVs form a new therapeutic agent class that induces anti-tumor immune responses by selectively killing tumor cells and inducing systemic anti-tumor immunity. On the contrary, several oncogenic viruses have been shown to play significant roles in malignancy progression in the case of gastric cancer. Therefore, this review evaluates the current state of research and advances in understanding the dual role of viruses in gastric cancer.

## 1. Introduction

The global burden of gastric cancer remains high. It is responsible for over 769,000 deaths (equating to 1 in every 13 deaths globally), ranking fifth for incidence and fourth for mortality globally [1]. Gastric carcinogenesis is a complex, multifaceted process primarily attributed to prolonged contact with pathogens; thus, the World Health Organization (WHO) describes gastric cancer as predominantly an infection-related malignancy. Therefore, an early diagnosis and preventative measures are imperative in managing gastric malignancy initiation in high-risk populations. Broadly, gastric cancer has two manifestations based on anatomical sites; cardia, limited to the upper stomach, and non-cardia, predominant in the mid to lower stomach [2]. These anatomically limited gastric cancer presentations differ in the risk factors, carcinogenesis, and epidemiologic patterns. The cardia gastric cancers are predominantly related to gastroesophageal reflux disease (GERD), resembling characteristics of esophageal adenocarcinoma (EAC) [3].

On the contrary, the non-cardia gastric cancers are attributed to the chronic mucosal inflammation caused by sustained infection, including but not limited to the bacterium, *Helicobacter pylori* (HP) [4], and onco-viruses such as Epstein–Barr virus (EBV), and hepatitis B virus (HBV). A meta-analysis in 2020 demonstrated gastric cancer patients with a significantly higher viral load of hepatitis C virus (HCV), human cytomegalovirus (HCMV), and human papillomavirus (HPV). However, the underlying causal relationship between infection of HBV, HCMV, HPV, and the risk of GC remained inconclusive [5]. Besides pathogenic infections, there are various risk factors for non-cardia gastric cancer, including alcohol consumption, tobacco smoking, saline-preserved foods, radiation exposure, sedentary lifestyles, obesity, and dietary adulterants [6]. On the flipside, OVs target and lyse the tumor cells while sparing the healthy cells [7]. Their mechanisms are multi-dimensional; they kill the tumor-supporting cells in the tumor microenvironment and expose the tumor-associated antigens with an antitumor immune response [7,8].

### 1.1. Gastric Cancers and Onco-Viruses

Viruses are pro-oncogenic and cause cancer in around one-tenth of the cases [9], such as; *EBV* (239,700–357,900 registered cases), HPV, associated with 85% of invasive cervical cancers, (ICC), HCV and HCB (associated nearly 20% hepatocellular carcinoma (HCC) cases in the west, and 60% HCC cases in Asia/Africa), reported as the most prevalent and frequently associated oncogenic viral pathogen [7,9].

Onco-viruses have evolved alongside their hosts and are not necessarily pathogenic [10]. They chronically persist at several human body sites by producing undetectable replicates of themselves. Therefore, evolutionarily the onco-viruses alone are not a driving factor to cause cancer so long as the host controls the operations [11]. The oncogenic potential of viruses is triggered by additional risk factors from the neighboring environment or the living host [11]. The onco-viruses also hijack the host’s “DNA methylation” system to camouflage an invasion, and the infection remains undetectable by the host’s methyl marker surveillance system [12,13]. Upon infecting the host, the viral DNA chromatinize and subtly maintains its DNA either; as a viral DNA insert into the host cell genome or independently, as an episome, a circular double-stranded DNA [14].

Virus-mediated tumorigenesis is a complex, multifactorial process. In addition to a viral entry, genetic and epigenetic changes transform normal, healthy cells into abnormal, tumor-producing cells, leading to aberrant cell signaling pathways favoring immortality [15,16,17]. The characteristic mutations determine the cellular interactions with the immediate microenvironment [15]. The viral oncogenesis is mediated by; the translation of viral oncoproteins such as E6, HPV, and E7 in the host cells that modify the host cellular interactome and transcriptome, and by the introduction of genetic mutations in the host cells that cause immune suppression in the tumor microenvironment (TME), inaccuracy in DNA repair, resistance to apoptosis, and the inactivation of host tumor suppressors [12,18,19,20].

In normal cells, pathogenic viral particles are detected and cleared by various signaling pathways stimulated through TLRs or by local interferon (IFN) release. The TLRs are pattern recognition receptors stimulated in response to repeated sequences unique to bacteria and viruses, such as pathogen-associated molecular patterns (PAMPs). The TLR pathway stimulates antiviral responses in host cells and promotes innate immunity through the downstream cellular factors such as TNF-associated factor 3 (TRAF3), IFN-related factor 3 (IRF3), IRF7, and retinoic acid-inducible gene 1 (RIG-1). These factors reinforce antiviral machinery through Janus kinase–signal transducer and activator of transcription (JAK–STAT) pathway resulting in local IFN release, which activates a protein kinase, protein kinase R (PKR) [21,22]. The viral activation of PKR terminates cell protein synthesis and promotes rapid cell death and viral clearance.

On the other hand, the cancer cells have a defective IFN pathway signaling and PKR activity, which interferes with viral clearance. Many viruses can also modulate signaling pathways such as WNT—catenin, Notch, Pi-3K-AKT, MAPK, -mTOR, and NK-B within tumor cells, preventing apoptosis and allowing the virus to complete its life cycle [19,20].

### 1.2. Epstein–Barr Virus Associated Gastric Cancer

EBV is a gamma herpes virus with a linear double-stranded DNA core enveloped by an icosahedral nucleocapsid and a tegument that infects either a stomach epithelial cell or a B lymphocyte cell. EBV infects more than 90% of the population worldwide and maintains a life-long latent phase of gene expression with intermittent lytic phases [23]. Most EBV genes are expressed during the lytic phase to facilitate genome replication, assembly, and production of viral particles [24]. Latent infection, however, minimizes the gene expression of latent proteins; (EBV-determined nuclear antigen 1 (EBNA1), 2, 3A, 3B, 3C, and EBNA-LP; latent membrane protein 1 (LMP1) and LMP2), noncoding RNA (EBER1 and EBER2), and viral miRNAs (BHRF1-miRNA and BART-miRNA), while simultaneously perpetuating the infection in the form of extrachromosomal circular DNA called episomes [25]. These episomes replicate alongside the host chromosomes within a small number of circulating host cells [24,26]. Several memory B-cells harbor a persistent life-long latent infection that can differentiate into plasma cells and re-enter the lytic EBV gene expression profile [27].

The EBV was the first virus associated with carcinogenesis and was identified from Burkitt’s lymphoma cell line in 1964 [24,26]. As a result, it has been extensively investigated in regard to different types of human cancers, including Hodgkin’s lymphoma, diffuse large B-cell lymphoma, and Burkitt’s lymphoma in immunocompromised individuals [28,29,30], oral hairy leukoplakia, CNS lymphoma, non-Hodgkin lymphoma, and lymphoproliferative disorders in immunocompromised hosts [31]. Some investigations also show the presence of EBV virus in lymph epigastric adenocarcinomas [32], lymphoepithelioma-like gastric carcinoma with a prominent lymphocytic stroma [33], and lymphoepigastric adenocarcinomas [32].

EBV-associated gastric carcinoma (EBVaGC) is a distinctive subset with 10 % accountability of all gastric malignancies [33,34]. Recent genome-wide molecular analysis conducted by ‘The Cancer Genome Atlas (TCGA)’ network suggests that EBV-associated gastric carcinoma forms a predominant class of gastric cancers and implicates that genetic and epigenetic alterations contribute to EBV progression associated gastric carcinogenesis [35]. A distinctive feature of the EBV-positive gastric carcinoma class is extensive hypermethylation of both promoter and non-promoter CPG islands [35,36,37]. It specifically hypermethylates the promoter of the CDKN2A gene but demethylates the MLH1, which is predominantly methylated in different subtypes of gastric cancers [35,38,39]. The extensive hypermethylation drive of both host and viral genome provides an apparatus for the virus to manipulate and control the fundamental cellular processes that promote viral persistence and propagation, as shown in Figure 1 [37,40,41,42,43,44].

TCGA analysis establishes that EBVaGC offers standard molecular GC features with a mutation in PIK3CA, amplification of JAK2, PD-L1 and PD-L2, ARID1A, and mutation in the BCOR gene [35]. Micro RNAs facilitate the malignant transformation of epithelial cells in EBV infection. EBV is known to encode its microRNA targeted against the genes controlling cellular processes such as apoptosis and simultaneously may alter the expression profile of host cellular micro-RNA.

### 1.3. Hepatitis B Virus and Gastric Cancers

HBV is a hepatotropic DNA virus that preferably infects hepatic cells and is estimated to cause chronic infection in 256 million cases worldwide. The infection with the HBV virus accounts for approximately 50% of hepatocellular carcinoma (HCC) cases worldwide [45,46]. However, increasing evidence suggests the involvement of HBV in the progression of extrahepatic carcinomas such as pancreatic cancers [47], colorectal cancers [48], and gastric carcinoma [49]. Oncogenic hepatitis B virus X protein (HBX) plays a central role in the progression of HBV-mediated hepatocellular carcinomas; however, in 2019, researchers found that HBX protein was also significantly higher in gastric carcinoma cells than in normal specimens [50]. In addition, hepatitis B viral proteins and genetic elements have been detected in non-hepatic tissues, suggesting that extrahepatic HBV infection might be sustained [50,51,52]. The underlying mechanism of gastric carcinogenesis through HBV infection remains elusive. However, co-morbidities such as chronic inflammation, systemic immune function override, liver cirrhosis, and direct impact of oncogenic HBV proteins in gastric cells may have a role to play [53,54].

Numerous independent investigations have established a link between gastric cancer and HBV infection by identifying HBV surface antigen on the gastric carcinoma cells over the last several years [50,51,52,55,56]. In 2015, a study confirmed the presence of the HBsAg serological antigen in gastric cancer patients [57]. The HBsAg antigen tested significantly among patients without any previous or positive family history of cancer (95 percent CI): (1.06–2.11). However, the most recent research, published in 2019, confirmed the link between HBV and gastric carcinoma [51]. The study investigated the association between HBV and gastric cancer in patients with and without *H. pylori* infection. According to the data, the HBsAg antigen was discovered in 83 (11.4%) of 728 patients, whereas the *H. pylori* infection was found in 408 (56%) individuals. In 69 patients, co-infection of *H. pylori* and HBV was detected (9.5 percent) [51]. Moreover, *H. pylori* infection was discovered substantially more often in individuals who tested positive for HBsAg than those who tested negative (*p* = 0.029) [51]. None of the patients infected with *H. pylori* and HBV had normal stomach tissue. This study confirms that HBV infection may be associated with the progression of precancerous lesions; however, it is possibly not sufficient to initiate gastric cancer. The hypothesis was supported by evidence that a combined HBV and *H. pylori* infection was found in many gastric cancer patients who died of the ailment [51]. Chronic inflammation may cause persistent transformations of the gastric epithelium, immune dysregulation, genetic instability, and epigenetic changes. HBsAg is an independent risk factor for liver cirrhosis [52], whereas cirrhosis is a risk factor for GC. Liver cirrhosis may cause hypoxia, a risk factor [55] for GC, and a poor prognosis for patients with GC.

### 1.4. Oncolytic Viruses

The OVs recognize, infect, and lyse the tumor cells, thereby reducing their burden [58]. The OVs such as H1 autonomous replication viruses are naturally tropic to tumor cells [58]. However, oncotherapy utilizes genetically engineered OVs to replicate in the tumor cells selectively [58,59,60,61,62,63,64]. The tumor microenvironment is linked intimately to the tumor core consisting of necrotic cells, hypoxic oxygenation levels, and acidic pH levels, primarily due to a limited vasculature. Furthermore, the immune system in this environment is abnormally regulated. In such conditions, tumorigenic cells survive with little to no immunological interference [65,66]. T-cell signals are blocked, and an immunologically privileged site of tumor proliferation results from the combination of neo-antigens, cytokines (e.g., TGFβ), and specialized regulatory cells (e.g., T-regs) within the TME [66,67,68]. The engineered OVs exhibit several mechanisms that direct the infected host cells to a lytic phase, leading to the apoptosis of the host’s tumor cell [7]. The lysis releases antigens into the surrounding tumor microenvironment that activate the host’s immune system and results in an antitumor/anti-viral response [69]. Therefore, OVs modify the tumor microenvironment from an unrecognizable to an antigenic state. It enables the host immune system to identify the abnormal cells that otherwise remain hidden and maintain a state of anticancer immunity [8,21].

## 2. General Mechanism

OVs exert their antitumor effect through; (a) the selective replication within the cancer cells that causes direct cell lysis within the TME and (b) through the induction of systemic immunity against tumors, later likely the most effective strategy [70,71,72]. The effectiveness of oncolysis depends on various factors, including the virus type, the dose, characteristics of the viral vector, natural and engineered tropism of the virus, and the interactions between the virus and tumor microenvironment. To maximize the specificity, the OVs are engineered to target cancer-specific molecular patterns such as; upregulated surface markers, [58,73,74,75], transcription factors [76,77,78], cancer-specific promotors, and intermediary metabolites [79,80]. Genetically engineered OVs that contain pro-apoptotic genes also called suicide gene elements such as TNF-related apoptosis-inducing ligand, TRAIL; TNFα; cytosine deaminase (CD); and adenovirus death protein, ADP), as a part of the molecular construct efficiently kill a cancer cell [69,81,82,83,84,85,86]. In several preclinical models, cancer cells specifically induced the expression of suicide genes using tumor-enriched or tissue-specific promoters to limit the side effects and improve therapeutic outcomes such as Ad-OC-HSV-TK driven by osteocalcin promoter [87].

Secondly, the OVs induce the systemic innate tumor-specific immunity by counteracting the cancer-mediated immune evasion. Oncolytic cancer cell death releases tumor-associated antigens also called neo-antigens, that stimulate an adaptive immune response in the tumor microenvironment. Viral molecules such as genetic elements and capsid proteins compose the pathogen-associated molecular patterns (PAMPs), and heat shock proteins, high mobility group box 1 (HMGB1) protein, calreticulin, ATP, and uric acid that compose cellular danger-associated molecular pattern signals (DAMPs) are released in the tumor microenvironment [88,89]. With the presence of danger signals (DAMP) and TLRs engagement, type I IFN levels and other inflammatory mediator levels increase, further augmenting the immune response against cancer. Thereby, cytokines such as type I IFNs, tumor necrosis factor-α (TNFα), IFNγ, and interleukin-12 (IL-12), maintain the inflamed environment around the site of antigen recognition by promoting the maturation of antigen-presenting cells (APCs) such as dendritic cells as shown in Figure 2 [90,91,92]. TNF-alpha stimulates tumor cell death through its antiangiogenic effects that destroy blood vessels supplying to the tumor [93]. These activate antigen-specific CD4+ and CD8+ T cell responses, which differentiate into cytotoxic effector cells capable of locating tumor sites, where they mediate antitumor immunity [94]. As part of the innate immune response, type I IFNs and DAMPs activate natural killer (NK) cells. The NK cells kill cancer cells with downregulated major histocompatibility complex (MHC) class I expression [95,96].

## 3. Clinical Implication of Oncolytic Viruses in Gastric Cancer

### 3.1. Herpes Simplex Virus

The herpes simplex virus-1 (HSV-1) is a double-stranded DNA virus that belongs to the alpha-herpesviruses subfamily [97,98]. The Food and Drug Administration (FDA) approved a non-pathogenic strain of HSV-1, Talimogene Laherparepvec (T-Vec), to treat metastatic melanoma [99,100]. The most studied oncolytic virus, T-VEC, was engineered by inserting GM-CSF in place of γ34.5 and ICP47 to inhibit the neurovirulence factors and initiate the viral replication and immunogenicity [101,102]. Upon introduction at the tumor site, these viruses aim to directly target the tumor by infecting the host tumor cells, replicating within the host, and consequently leading to the lysis of the infected tumor cell [103]. The release of tumor neo-antigens into the surrounding milieu stimulates the immune system and promotes a heightened inflammatory response [103]. Additionally, in vitro analysis of two second-generation genetically modified oncolytic herpes simplex viruses, NV1020 and G207, demonstrate an oncolytic effect on gastric cancer cells [104]. Applying a mouse xenograft model of peritoneally disseminated gastric cancer indicated NV1020 was more efficient than G207 at lower viral doses. However, it required an intraperitoneal treatment of the oncolytic virus for a positive impact [104]. Moreover, a combinatorial administration of G207 with mitomycin C (MMC) demonstrated considerable synergism against the gastric cancer cells [104]. The third generation of oncolytic HSV-1, G47Δ, is considered a novel and attractive therapeutic approach for solid tumors. In vitro administration of G47* had a satisfactory proliferative and cytotoxic impact on several human GC cell lines studied. Additionally, intratumorally administration of G47* resulted in an increase in the expression of the immunostimulatory molecule (soluble CD80) and IL-12 and enhanced M1 macrophages polarization and infiltration in vivo, which inhibited the growth of subcutaneous tumors. G47* treatment was also associated with an increase in cytotoxic NK cells [105]. Oncolytic HSV can be enhanced by producing an HSV expressing TSP-1, along with anti-angiogenic effects on tumor cells. In vitro and in vivo studies were conducted with a third-generation oncological HSV (T-TSP-1) expressing human TSP-1 [106]. In vivo administration of TSP-1 resulted in oncolysis in addition to the inhibition of angiogenesis by suppressing the TGF-* signaling pathway [106]. Researchers demonstrated enhanced cytotoxicity in MKN45, MKN28, and MKN1 cells in vitro when using fourth-generation oncolytic HSVs that contain the ICP6 gene regulated by the hTERT promoter. These findings indicate that the use of therapeutic HSVs with the ICP6 gene under the control of the hTERT promoter may be beneficial and effective for the treatment of GC [107]. The results from a third-generation HSV oncolytic suppressor of cytokine signaling 3 (SOCS3) showed an increase in proliferation and tumor cell lysis properties for the MKN1 cell line as well as in human GC specimens [108].

### 3.2. Virus of Vesicular Stomatitis

Vesicular stomatitis virus (VSV) belongs to a Rhabdoviridae family that replicates and triggers apoptosis in various cell types, including cancer cells. However, VSVs show anti-cancerous function in gastric cancer [109]. The VSV matrix protein (MP) expression in gastric carcinoma cell line MKN28 triggered apoptosis and limited its proliferation [109].

### 3.3. Virus Vaccinia

The enveloped double-stranded DNA Vaccinia virus of the Poxviridae family [110] may be a promising candidate for gastric carcinoma therapy for which clinical trials (phase I and II) have been completed (NCT01443260). The bulk of Vaccinia virus (VV) particles are mature intracellular virions produced from a single lipid bilayer envelope that remains confined chiefly within the infected cell until lysis [111]. The other two infectious species, cell-associated enveloped viruses (CAEV) and extracellular enveloped viruses (EEV) contain an additional lipid bilayer and bud out from the host cell without lysing it [112]. The vaccinia virus can absorb enormous quantities of foreign DNA while maintaining high safety and replication efficiency in humans [64]. GLV-1 h153, a genetically altered vaccinia virus that carries the human sodium iodide symporter (hNIS) gene, is explored as a potentially novel treatment for GC. The GLV-1 j153 has shown a promising oncolytic effect with over 90% cytotoxicity in five human gastric cancer lines [64]. The cytotoxicity may be enhanced by combined therapy of radioiodine and GLV-1 h153; however, it remains to be investigated further [64].

## 4. Conclusions

OVs evolve as a way of bypassing the immune evasion mechanisms of the tumor, proposing to amend the clinical manifestation of patients by stimulating the host immune system or by the direct lysis of tumor cells. The present-day genetic engineering techniques have provided ways to improve the production of safe and efficient OVs, targeting the virus to the tumor, and decreasing the adverse effects of their use. Oncolytic virotherapy is unquestionably one of the options as some viruses such as HSV, VSV, and VV possess oncolytic properties; these viruses seem a better option in cases where radiotherapy, and chemotherapy fails. Furthermore, it is possible to observe significant effects of the clinical use of OVs, whether in single or combination therapy, to treat tumors. This review evaluates the current state of research and advances in deepening our knowledge of the dual roles of viruses in gastric cancer. The underlying mechanism of these viruses in disease progression and treatment in gastric cancer is still debatable and needs further research. However, oncolytic virotherapy is gaining much attention as one of the possible treatment options for gastric cancer.

## Figures and Tables

**Figure 1 vaccines-10-00600-f001:**
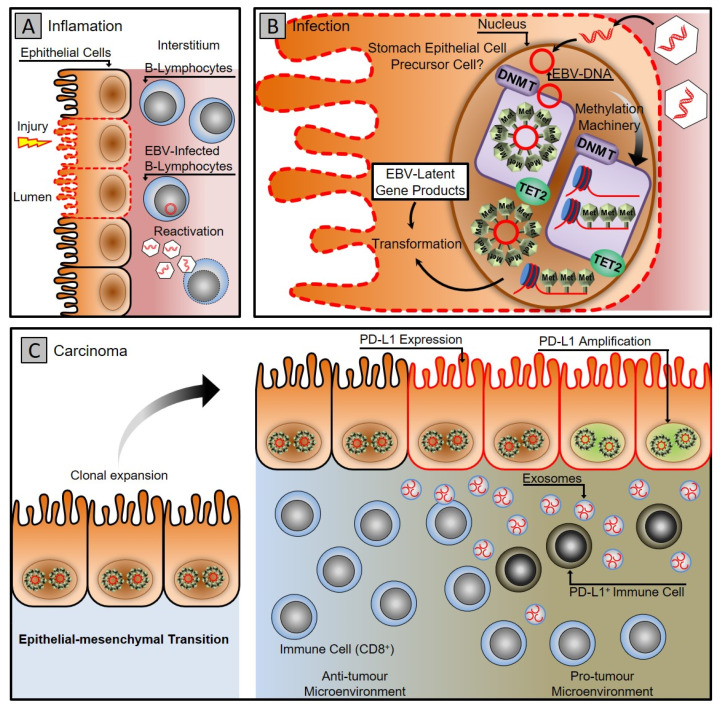
Gastric carcinoma associated with Epstein–Barr virus infection. (**A**) Gastritis stage: latent EBV DNA is recruited to the stomach mucosa, infecting epithelial cells. (**B**) Infection stage: A latent infection is established in the nucleus of the epithelial cell by EBV. The DNA methylation machinery is activated, turning infected cells into clones. (**C**) Carcinoma stage: the virus uses cellular machinery to manipulate cells and the microenvironment while counteracting the host immune system using exosomes. To evade the host immune system, cancer cells express PD-L1 and recruit PD-L1-positive immune cells.

**Figure 2 vaccines-10-00600-f002:**
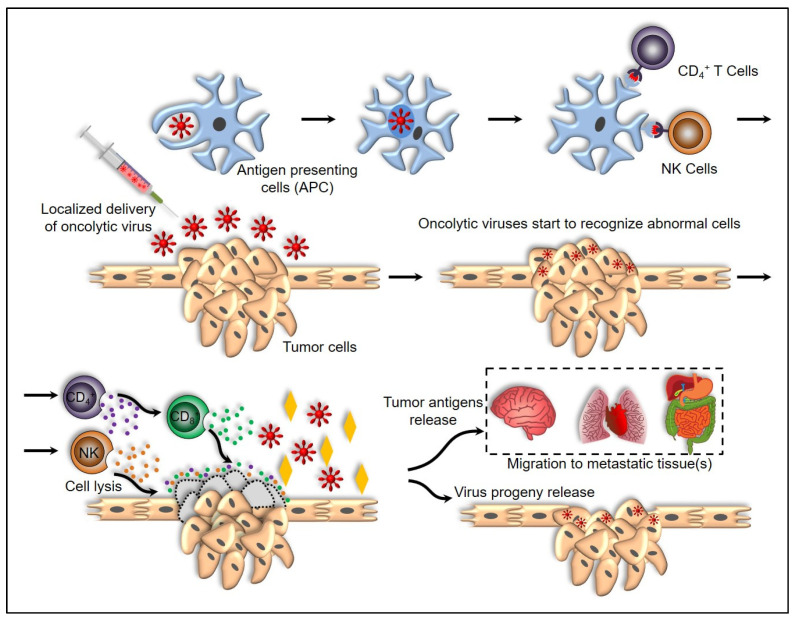
Oncolytic virus mechanism of action. Initial administration of OVs can occur intravenously, subcutaneously, intraperitoneally, and intrathecally. A combination of natural tropism and genetic targeting prefers the entrance of OVs into tumor cells. Later, these viruses start recognizing tumor cells and infect host cells via connection by different receptors and substances present in the tumor environment. After this point, viral replication starts using the cellular machinery, leading to the formation of viral proteins, a reduction in functioning of cell, state of oxidative stress, and initiation of pathways associated with autophagy. viruses are enclosed by APCs that form endosomal vesicles which attach with lysosomes to digest them into smaller particles inside the cell. A favorable environment results from expressing the class 2 proteins of the major histocompatibility complex on the surface of infected cells, which further stimulates and activates T cells, involves cytokine production, and directs action on the infected cells. Through viral action and immune response, tumor cells are destroyed, releasing the virus progeny inside the host. This enables the virus to infect other tumor cells and combat the tumor. Finally, a new type of inflammatory response can be triggered by cell death, as it expresses tumor antigens that can be detected by the immune system, in turn targeting the surrounding tumor and metastatic sites.

## Data Availability

Not applicable.

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
