# Peer review of "Gastric Cancer and Viruses: A Fine Line between Friend or Foe"

_vaccines, 2022, doi:10.3390/vaccines10040600_

Round 1
Reviewer 1 Report
Firoz et al presented a review article about viruses associated with gastric cancers (GC) and the possible mechanisms of GC. The review articles includes EBV, HBV and oncolytic viruses.
The reviews lacks a lot of information about viruses associated with GC such as HCV, CMV, JCV,...etc. I would suggest the authors to read the new published metanalysis which covers most of viruses associated with GC
https://www.ncbi.nlm.nih.gov/pmc/articles/PMC7386361/
Major points
1- The review needs extensive language editing
2- As mentioned before, a lot of missing information about viruses associated with GC should be included.
3-
4-References styles are not consistent. The reference follow numbering system, however in text there are other styles of referencing. For example, page 3 , lines 116-119 "Furthermore, increasing evidence suggests the involvement of HBV in the progression of extrahepatic carcinomas such as pancreatic cancers [35], colorectal cancers (Su et al., 2021), and gastric carcinoma [37].
There are many examples for different referencing style.
5- The authors should include list of abbreviations, and the abbreviated name should be mentioned in the first time it appears in text.
6- Scientifically, some sentences need to be revised. For example; page 5 lines 190-195 "The pattern recognition receptors recognize these molecular patterns (PAMPS), toll-like receptors (TLRs) expressed by the dendritic cells and stimulate the production of antiviral inflammatory molecules such as type 1 interferons, tumor necrosis factor-alpha (TNF-alpha), and cytokines such as interleukin 2 (IL-2), which maintain the inflamed environment around the site of antigen recognition [55].
type I interferon, TNF-a, IL2 all of them are cytokines, not only IL-2 is cytokines and other molecules are not.
The authors need to revise the manuscript carefully
Reviewer 2 Report
Thank you for sending me the research article paper “Viruses and their underlying mechanism in gastric cancer: An 2 update” for review in the Vaccines. In the article of Ahmad et al., the author discussed the role of viruses in the development of gastric cancer. There are important points that should be discussed and improved.
- Author should write an abstract in a more precise way. Title focuses on the role of viruses but the abstract presents another way of literature. Author should discuss more about the virus and its role in the development of cancer. Why do authors think that viruses are important?
- I think the author should also focus on the role of H. pylori in a separate paragraph. As the author mentioned in the introduction. Also change the title.
- Along with the H. pylori, there is also a need to discuss other gastric cancer causing factors, including high salt, capsaicin, stress and alcohol consumption and play a role in the gastric cancer development. As the author mentioned in the introduction part. (Prepare a separate paragraph and discuss H. pylori and other lifestyle, dietary and host factors).
- There is a need to discuss other virulence factors of H. pylori, like flagella, pili, catalase, and oxidase as well as discuss blood group antigen like LeY antigen and post-translational modification enzyme (FUT1/FUT4) and its role in the CagA. Further there is need to discuss the less importance than CagA toxin to target for drug.
- It would be significant to present the manuscript in the form of a table. e.g. different types of virus in the development of gastric cancer, mechanism, treatment, experiment type.
- It would be a good idea to present the review in a format like this, Introduction of virus associated gastric cancer, mechanism of specific virus, possible treatment and role of other factors. (heading: 1.1, 1.2, 1.3, 1,4, and heading 2).
Round 2
Reviewer 1 Report
The authors addressed my comments
Reviewer 2 Report
Accepted